# Simplified Selection Criteria for Secondary Cytoreductive Surgery in Recurrent Ovarian Cancer

**DOI:** 10.3390/cancers14163987

**Published:** 2022-08-18

**Authors:** Joo-Hyuk Son, Jimin Lee, Sun-Hyung Yum, Jeeyeon Kim, Tae-Wook Kong, Suk-Joon Chang, Hee-Sug Ryu

**Affiliations:** 1Division of Gynecologic Oncology, Department of Obstetrics and Gynecology, Ajou University School of Medicine, 164 Worldcup-ro, Yeongtong-gu, Suwon 16499, Korea; 2Department of Obstetrics and Gynecology, Ajou University School of Medicine, Suwon 16499, Korea

**Keywords:** advanced ovarian cancer, recurrent ovarian cancer, secondary cytoreductive surgery

## Abstract

**Simple Summary:**

Eligibility for secondary cytoreductive surgery (SCS) in ovarian cancer is dependent on multiple confounding factors. In this study, we evaluated the clinical characteristics of 262 patients with recurrent ovarian cancer to assess the impact of SCS on patient survival and establish simplified criteria for the selection of patients who would most likely be benefitted from SCS. We observed that the median survival was significantly longer in the patients who received SCS compared with those who received chemotherapy alone. As for the indication of the surgery, limited regional recurrence (single region or up to three regions with limited carcinomatosis) emerged as the simplified factor that could predict no residual disease after SCS.

**Abstract:**

(1) Background: Multiple confounding factors influence the indications for secondary cytoreductive surgery (SCS) in patients with ovarian cancer (OC). We aimed to identify the factors associated with patients most likely to benefit from SCS. (2) Methods: We retrospectively reviewed the medical records of patients with recurrent ovarian cancer from 2003 to 2021. The potential factors influencing treatment outcomes and survival between patients who received chemotherapy alone and those who received SCS after recurrence were evaluated. (3) Results: Recurrent OC was identified in 262 patients, with a median age of 53 (20–80) years. Of these patients, 87.4% had an initial stage III/IV disease. Eighty-nine (34%) patients received SCS. The median survival was 41.0 (95% confidence interval [CI], 37.4–44.5) months and 88.0 (95% CI, 64.2–111.7) months in the chemotherapy and surgery groups, respectively. A multivariate analysis showed limited regional carcinomatosis (single region or up to three regions with limited carcinomatosis) (*p* = 0.045) as the only significant factor for predicting no residual disease after SCS. In platinum-sensitive recurrent patients with limited regional recurrence, the complete resection rate was 87.6%. (4) Conclusions: SCS had a significant impact on survival in the selected patient population. Limited regional recurrence (single region or up to three regions with limited carcinomatosis) may be a simple criterion for SCS in platinum-sensitive recurrent OC patients.

## 1. Introduction

Ovarian cancer is the leading cause of death from gynecological cancer worldwide [1]. Despite an initial successful response to primary therapy, about 80% of the patients experience disease recurrence [2,3]. To date, numerous retrospective data supported the role of secondary cytoreductive surgery (SCS) in select patients with recurrent ovarian cancer [3,4,5]. In several meta-analyses, complete tumor debulking, resulting in no gross residual disease, has been associated with overall survival benefits [6,7]. Nonetheless, its beneficial impact on survival outcomes remains controversial, even in recent randomized controlled trials. The GOG (Gynecologic Oncology Group)-213 trial failed to show the superiority of the surgery, whereas the AGO-Desktop III and SOC-1 trials showed promising results for SCS in well-selected, platinum-sensitive, recurrent ovarian cancer patients [8,9,10]. There might be several reasons for these contradictory results. Among them, the most important factor may be the patient selection strategy [11]. In the GOG-213 trial, eligibility was based on the surgeons’ preferences, without uniform selection criteria across the participating centers. However, the DESKTOP III and SOC-1 trials used strictly defined criteria. The SOC-1 trial used a scoring system based on six parameters: the International Federation of Gynecology and Obstetrics (FIGO) stage at initial diagnosis (stage III/IV, 0.8), any residual disease after primary cytoreduction (>0 cm, 1.5), a progression-free interval (<16 months, 2.4), the performance status (Eastern Cooperative Oncology Group [ECOG] 2–3, 2.4), any serum cancer antigen (CA)-125 at recurrence (>105 U/mL, 1.8), and any ascites at recurrence (present, 3.0). Patients with total scores of 0–4.7 were categorized in the low-risk group for residual disease after a secondary cytoreductive surgery. The DESKTOP trial used the Arbeitsgemeinschaft Gynaekologische Onkologie (AGO) score, which includes a patient’s performance status, ascites, and residual disease status after the primary debulking surgery. Considering that complete resection has been the most significant prognostic factor to date, a correct selection of patients who will benefit from the surgery is paramount.

Currently, several selection criteria have been developed to predict the likelihood of a complete resection in patients with recurrent ovarian cancer [12,13,14]. Tian et al. developed a model based on individual data from 1075 patients with recurrent ovarian cancer [14]. In their study, complete secondary cytoreduction was achieved in 53% of patients with a low score (0–4.7) and 20% of patients in the high-risk group (>4.7). The DESKTOP OVAR trial (Descriptive Evaluation of preoperative Selection KriTeria for OPerability in recurrent OVARian Cancer) developed an Arbeitsgemeinschaft Gynaekologische Onkologie (AGO) score for predicting a complete gross resection (R0) after SCS. A positive AGO score was characterized by an ECOG performance status of 0, ascites of <500 mL, and a complete resection at initial surgery [13]. The Memorial Sloan Kettering Cancer Center (MSKCC) group recommended SCS based on the combination of disease-free survival and number of recurrent sites [12].

Although these criteria help to identify patients with a high probability of successful surgery, complete resection rates vary widely due to variations in the criteria. In addition, as reported previously, these known criteria might be strict, which may sometimes prohibit beneficial treatment for patients who do not meet these criteria [15]. In this study, we aimed to evaluate the clinical outcomes of recurrent ovarian cancer and investigate simplified selection criteria to identify patients who may benefit from the surgery.

## 2. Materials and Methods

Patients with recurrent ovarian cancer who were treated at Ajou University Hospital, Suwon, South Korea, from May 2003 to April 2021 were identified. The key inclusion criteria were as follows: (i) patient age > 18 years; (ii) a diagnosis of platinum-sensitive recurrent epithelial ovarian cancer; and (iii) good performance status (ECOG 0–1). Patients with non-epithelioid histology or patients who did not receive standard chemotherapy or surgery were excluded (Figure 1). The study protocol was approved by the Institutional Review Board of Ajou University Hospital (AJIRB-MED-MDB-21-086). As a retrospective study, the need for informed consent was waived by the IRB. All patients were surgically staged and received paclitaxel/carboplatin-based adjuvant chemotherapy. (The recurrence of the disease was clinically defined primarily using tumor markers, computed tomography (CT), or positron emission tomography scans. As factors for SCS, the following aspects were considered: residual disease status after primary cytoreductive surgery, progression-free survival (PFS), performance status, site of recurrent disease, ascites, and, most importantly, expected residual disease after the second surgery.

If a patient was not suitable for SCS, chemotherapeutic agents were selected following the National Comprehensive Cancer Network’s guidelines for platinum sensitivity. The SCS was defined as a surgical procedure performed at some time (with a disease-free interval of more than 6 months) after the completion of primary therapy with the purpose of tumor cytoreduction [16,17]. In this regard, patients who received a cytoreductive surgery after second- or third-line chemotherapy also included in the SCS group. To clarify the characteristics of recurrent ovarian cancer, the following factors were analyzed: age at diagnosis, FIGO stage, type of primary surgery, residual disease status after primary surgery, histology, breast cancer gene (BRCA) mutation status, CA-125 at recurrence, and recurrent site. There was a wide variety of recurrent sites, such as peritoneum; liver; spleen; intestines; retroperitoneal lymph nodes; and extra-abdominal sites, such as chest, brain, bone, and abdominal wall. For proper analysis, we classified them as limited regional carcinomatosis, extra-abdominal disease, and multiple lesions with diffuse carcinomatosis. Limited regional carcinomatosis included single lesion, either intra- or extra-abdominal; multiple intra-abdominal lesions (up to 3 sites) without diffuse peritoneal carcinomatosis; and limited carcinomatosis, such as localized peritoneal metastasis (e.g., right diaphragm peritoneum, paracolic gutter, and pelvic peritoneum). An analysis of the recurrent sites was based on standard CT scan results. Next, the treatment outcomes were comparatively analyzed between patients who received chemotherapy only (chemotherapy group) and those who received SCS (surgery group) for disease recurrence. The prognostic factors for the overall survival rate for patients who received SCS were then evaluated. Using these factors, the selection criteria for the SCS were identified.

The patient population and type of intestinal surgery were described using descriptive statistics. The Mann–Whitney test or chi-squared test was used to compare treatment outcomes between the two groups. The Kaplan–Meier method was used to perform survival analyses. The time from the date of initial diagnosis to the date of death by any cause was defined as the overall survival (OS). Progression-free survival (PFS) was defined as the period from the date of primary surgery to the first observation of disease progression. The statistical analysis was performed with IBM SPSS Statistics for Windows (version 25.0, IBM Corp., Armonk, NY, USA), with *p* < 0.05 defined as statistically significant.

## 3. Results

### 3.1. Patients’ Characteristics

A total of 262 patients with recurrent ovarian cancer were identified during the study period. The patients’ median age was 53 years. Most patients (87.4%) had initial FIGO stage IIIC–IV disease, and 72.1% of the patients received primary debulking surgery. Approximately half of the recurrent patients had gross residual (GR) disease after primary treatment (GR-1, *n* = 71 [27.1%]; GR-B, *n* = 58 [22.1%]). Of all patients, 89 (34.0%) received SCS, and 173 (66.0%) received only chemotherapy (Table 1).

### 3.2. Comparative Analysis of the Treatment Groups

Patients in the surgery group were more likely to be young and have an early FIGO stage, no GR disease after primary surgery, a higher rate of BRCA mutation, limited carcinomatosis, longer PFS (median, 19 months), and a lower rate of ascites. Patients in the surgery group showed significant survival gain compared with those in the chemotherapy group (Figure 2, Table 2).

### 3.3. Predicting Factors for Complete Resection after SCS

Among patients with a good prognosis (PFS > 12 months, *n* = 27), 26 (96.2%) with single regional recurrence showed a complete tumor resection, and 1 (3.8%) had GR disease < 1 cm. Conversely, in patients who had multiple lesions with limited carcinomatosis (*n* = 40), 67.5% (*n* = 27) had a complete tumor resection. To find factors related to the complete resection in SCS, a multivariate logistic regression was performed. A metastatic site (limited regional recurrence) was the only significant factor for predicting a complete resection after SCS (*p* = 0.045; Table 3).

### 3.4. Analysis with Known Selection Criteria

Patients’ inclusion rates and complete resection rates were evaluated among several selection criteria. When adopting the AGO criteria (ECOG, no residual disease in primary debulking surgery, and no ascites) for this study cohort, the inclusion and complete resection rates were 70.8% and 76.2%, respectively. The MSKCC criteria (PFS and recurrent sites) included 66.3% of the patients, and 74.6% of the patients had a complete resection. The criteria based on the significant factors in this study, which included a PFS > 12 months and limited carcinomatosis at recurrence, showed an inclusion rate of 74.1% and a complete resection rate of 78.8%. When adopting the criteria of a PFS > 6 months with limited carcinomatosis at recurrence, the inclusion rate was 100%, and the complete resection rate was 87.6% (Table 4).

## 4. Discussion

Our study primarily investigated simple clinical factors that could identify patients who would benefit from SCS when compared with previously known criteria. Based on this study, even with residual disease after primary surgery or ascites at the time of recurrence, a complete resection could be obtained in a well-selected patient population, suggesting that the SCS should not be determined based on the known criteria alone. The data suggest that limited regional carcinomatosis (single or up to three sites with limited carcinomatosis) may be used as a simplified criterion for platinum-sensitive recurrent ovarian cancer patients.

Indications for SCS in ovarian cancer are often dependent on multiple confounding factors. Currently, several selection criteria are available that predict the likelihood of a complete resection in patients with recurrent ovarian cancer. In 2006, Chi et al. reported the guidelines and selection criteria for recurrent ovarian cancer surgery based on 153 patients (from 1987 to 2001) in the Memorial Sloan Kettering Cancer Center [12]. These criteria were based on the site of recurrence (i.e., single, multiple, and carcinomatosis) and disease-free interval (DFI). If patients had a single site recurrence, SCS was offered if their DFI was >6 months. If patients had a DFI > 30 months, SCS was offered regardless of the recurrence site. If patients had multiple site recurrences or carcinomatosis, the decision might be individualized based on the DFI, age, and performance status. The complete resection rate was 41% in their entire cohort. A series of AGO-DESKTOP OVAR trials on surgery for recurrent ovarian cancer occurred contemporaneously with the MSK study [13]. In contrast with the MSK criteria, a good performance status, an absence of ascites, and the outcome of the primary surgery/initial FIGO (International Federation of Gynecology and Obstetrics, London, UK) stage comprised the AGO score. The score model was subsequently verified to positively predict surgical outcomes in a prospective multicenter trial (DESKTOP II trial) of 516 patients with recurrent ovarian with a complete resection rate of 76% [18]. Tian et al. reported that complete secondary cytoreduction was associated with six variables: the FIGO stage (odds ratio [OR] = 1.32, 95% confidence interval [95% CI]: 0.97–1.80), residual disease after primary cytoreduction (OR = 1.69, 95% CI: 1.26–2.27), PFS (OR = 2.27, 95% CI: 1.71–3.01), ECOG performance status (OR = 2.23, 95% CI: 1.45–3.44), CA-125 (OR = 1.85, 95% CI: 1.41–2.44), and ascites at recurrence (OR = 2.79, 95% CI: 1.88–4.13) [14]. They suggested a scoring system ranging from 0 to 11.9, and patients with a score of 0–4.7 were categorized as the low-risk group. The complete cytoreduction rate of the low-risk group was reported as 53.4%, compared with 20.1% in the high-risk group. A few studies conducted some exploration using positron emission tomography–laparoscopy to select suitable patients with ROC for successful SCS [19]. In a study of an innovative method using artificial intelligence (AI), three main factors—DFI, retroperitoneal recurrence (importance = 0.178), and RD at primary surgical treatment (importance = 0.138)—were suggested to predict a complete resection using an artificial neuronal network analysis [20]. However, these predictors have not yet been modeled and lack validity.

Though these criteria are beneficial for patient selection, their applicability may be limited due to wide variations and high stringency, which may prohibit patients who do not meet these criteria from receiving SCS [15]. In our study, all patients had a PFS > 6 months and limited regional carcinomatosis with single or multiple lesions (no diffuse-carcinomatosis in preoperative evaluation). Among the patients with ascites (*n* = 5), 60% (*n* = 3) received a complete resection. Additionally, among the patients with residual disease at primary surgery (*n* = 24), 79.2% obtained a complete resection after SCS. Although a previous study reported that residual disease was a risk factor for the GR disease after SCS, in patients who have a relatively long PFS with good chemo-response, residual disease at primary surgery might not be a risk factor (median PFS was 15 months in the complete resection group vs. 10 months in the GR group after SCS). In addition, despite ascites being widely known as a risk factor, our observations showed that a recurrent site was the most significant factor in the multivariate analysis. Therefore, we suggest that SCS should not be determined (or denied) based on the known criteria alone.

In this study, 23 patients (25.8%) in the surgery group had unresectable extra-abdominal disease at the time of the first recurrence. After receiving second- to fourth-line chemotherapy, the lesion had diminished in size, and the patients were included as per the eligibility criteria (PFS > 12 months and limited regional recurrence) for SCS. Like neoadjuvant chemotherapy before interval debulking surgery, additional second- to fourth-line chemotherapy was used to achieve a maximum resection in SCS. The complete resection rate in patients with extra-abdominal disease was 78.3% (18/23), and the second median PFS after SCS was 13 months (95% CI, 8.0–17.9 months). These findings suggest that if chemotherapies are adopted properly in the selection, the inclusion rate can be widened to include patients not eligible at the time of first recurrence and patients with extra-abdominal diseases.

This study has several limitations due to the retrospective design and relatively small number of patients. Since the surgical outcomes are mainly dependent on proper patient selection, surgical outcomes may have a risk of selection bias away from the null. Therefore, a sample size that is at least 10 times the number of predictors should be used for the model in order to achieve sufficient power in a multivariate logistic regression analysis [21]. The sample size in our study exceeded this minimum benchmark. The analysis contained eight predictors, and the study included 89 cases in the surgical group. Although this sample size sufficiently exceeded the minimum benchmark, our conclusions should be considered provisional and need to be supported by a study with a larger patient population. Even with these limitations, our study suggested simplified criteria usable in the practice of selecting patients with recurrent ovarian cancer. Since the surgery was mainly done with a single surgeon in a single center, the quality of surgery was qualified during the study period.

## 5. Conclusions

In conclusion, SCS showed a significant survival impact on a well-selected patient population. Limited regional recurrence (single region or up to three regions with limited carcinomatosis) may be used as a simplified criterion for the SCS. If chemotherapy is adopted properly, the patient selection can be widened and even include patients who were not eligible at the time of first recurrence or those accompanied by extra-abdominal diseases.

## Figures and Tables

**Figure 1 cancers-14-03987-f001:**
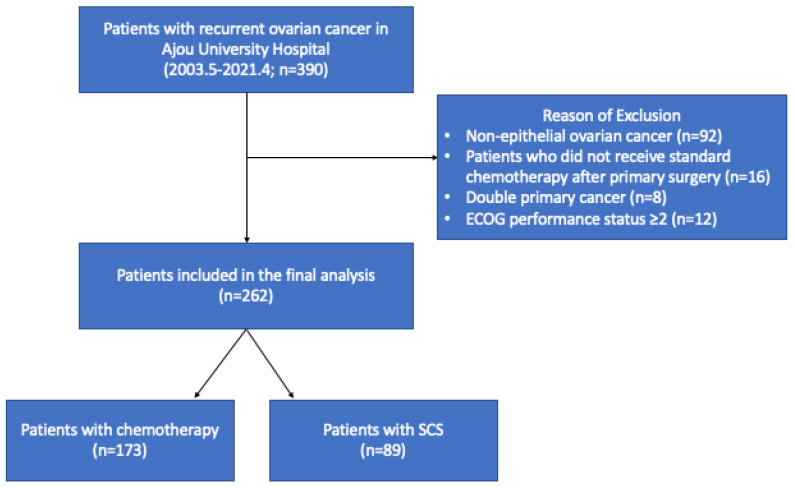
Flowchart of the patients in the study.

**Figure 2 cancers-14-03987-f002:**
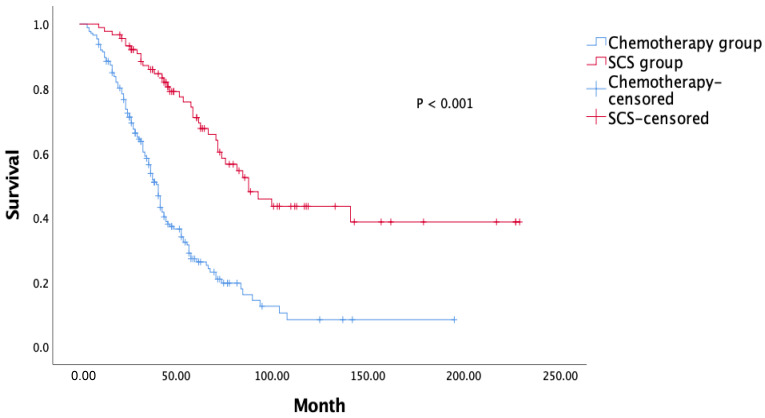
Overall survival analysis.

**Table 1 cancers-14-03987-t001:** Characteristics of patients with recurrent ovarian cancer (*n* = 262).

Age, years	53 (20–80)
Initial FIGO stage	
stage I–II	33 (12.6%)
stage III–IV	229 (87.4%)
Type of primary surgery	
PDS	189 (72.1%)
IDS	73 (27.9%)
Residual disease status	
NGR	131 (50.0%)
GR-1	71 (27.1%)
GR-B	58 (22.1%)
Unknown	2 (0.8%)
Histology	
Serous	224 (85.5%)
Non-serous	38 (14.5%)
*BRCA* mutation	
*BRCA1* mutation	12 (4.6%)
*BRCA2* mutation	4 (1.5%)
VUS	15 (5.7%)
Normal	66 (25.2%)
Unknown (not performed)	165 (63.0%)
Characteristics of recurrence	
CA-125 (U/mL)	69.3 (1.4–5770)
Limited carcinomatosis	45 (17.2%)
Ascites	30 (11.5%)
Extra-abdominal disease	73 (27.8%)
Chest	25 (9.5%)
Brain	6 (2.3%)
Bone	4 (1.5%)
Extra-abdominal LNs	33 (12.6%)
Abdominal wall	5 (1.9%)
Multiple lesions or diffuse carcinomatosis	162 (61.8%)
Treatment for recurrent disease	
SCS	89 (34.0%)
Chemotherapy	173 (66.0%)
PFS, months	15 (13.7–16.2)
OS, months	53.0 (45.2–60.7)

Results are expressed as median (95% confidence interval) or number (%). FIGO, The International Federation of Gynecology and Obstetrics; PDS, primary debulking surgery; IDS, interval debulking surgery; NGR, no gross residual disease; GR-1, gross residual disease size < 1 cm in the maximal diameter; GR-B, gross residual-bulky; *BRCA*, breast cancer gene; VUS, variant of uncertain significance; CA-125, cancer antigen 125; LN, lymph nodes; SCS, secondary cytoreductive surgery; PFS, progression-free survival; OS, overall survival.

**Table 2 cancers-14-03987-t002:** Comparative analysis of the treatment groups.

	Chemotherapy (*n* = 173)	SCS (*n* = 89)	*p*-Value
Age, years	55 (25–80)	50 (20–78)	0.001
Initial FIGO stage			0.001
stage I–II	12 (6.9%)	21 (23.6%)	
stage III–IV	161 (93.1%)	68 (76.4%)	
Residual disease at primary surgery			0.001
NGR	66 (38.2%)	65 (73%)	
GR-1	54 (31.2%)	17 (19.1%)	
GR-B	52 (30.1%)	6 (6.7%)	
Unknown	1 (0.6%)	1 (1.1%)	
Histology			0.462
Serous	150 (86.7%)	74 (83.1%)	
Non-serous	23 (13.3%)	15 (16.9%)	
*BRCA* mutation			0.001
*BRCA1* mutation	2 (1.2%)	10 (11.2%)	
*BRCA2* mutation	1 (0.6%)	3 (3.4%)	
VUS	11 (6.4%)	4 (4.5%)	
Wild type	35 (20.3%)	31 (34.8%)	
Unknown	123 (71.5%)	41 (46.1%)	
Characteristics of recurrence			
Limited carcinomatosis	11 (6.9%)	34 (39.1%)	0.001
Ascites	25 (14.5%)	5 (5.6%)	0.04
Extra-abdominal disease	59 (34.1%)	23 (25.8%)	0.172
CA-125 (U/mL)	114.7 (1.4–5770)	39.1 (1.4–2998.5)	0.108
PFS, months	14 (12.6–15.3)	19 (16.5–21.4)	0.001
OS, months	41 (37.4–44.5)	88 (64.2–111.7)	0.001

Results are expressed as median (95% confidence interval) or number (%). FIGO, The International Federation of Gynecology and Obstetrics; PDS, primary debulking surgery; IDS, interval debulking surgery; NGR, no gross residual disease; GR-1, gross residual disease size < 1 cm in the maximal diameter; GR-B, gross residual-bulky; *BRCA*, breast cancer gene; VUS, variant of uncertain significance; CA-125, cancer antigen 125; LN, lymph nodes; SCS, secondary cytoreductive surgery; PFS, progression-free survival; OS, overall survival.

**Table 3 cancers-14-03987-t003:** Multivariate logistic regression analysis to predict residual disease after SCS.

	Adjusted HR	95% CI	*p*-Value
Age	1.021	0.967–1.078	0.447
FIGO stage (Stage I/II vs. III/IV)	0.412	0.116–1.462	0.17
Any residual disease at the time of primary surgery	0.442	0.120–1.626	0.219
PFS > 12 month	0.559	0.168–1.843	0.337
*BRCA* mutation	0.325	0.036–2.952	0.318
Limited regional recurrence	0.259	0.069–0.968	0.045
Ascites before SCS	2.169	0.300–15.708	0.443
Extra-abdominal disease	0.675	0.184–2.477	0.554

SCS, secondary cytoreductive surgery; FIGO, The International Federation of Gynecology and Obstetrics; HR, hazard ratio; CI, confidence interval; PDS, primary cytoreductive surgery; BRCA, breast cancer gene.

**Table 4 cancers-14-03987-t004:** Analysis with known selection criteria and proposed criteria (Ajou criteria) for SCS.

	Inclusion Rate (*n*, %)	Complete Resection Rate (*n*, %)
AGO criteria	63 (70.8%)	48 (76.2%)
MSKCC	59 (66.3%)	44 (74.6%)
Tian criteria (low risk)	82 (92.1%)	64 (78%)
Ajou criteria		
PFS > 12 + limited regional recurrence	66 (74.1%)	52 (78.8%)
PFS > 6 + limited regional recurrence	89 (100%)	78 (87.6%)

AGO, Arbeitsgemeinschaft Gynaekologische Onkologie; MSKCC, Memorial Sloan Kettering Cancer Center; PFS, progression-free survival.

## Data Availability

The data that support the findings of this study are available from the corresponding author, upon reasonable request.

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
