# Peer review of "Simplified Selection Criteria for Secondary Cytoreductive Surgery in Recurrent Ovarian Cancer"

_cancers, 2022, doi:10.3390/cancers14163987_

Round 1

Reviewer 1 Report

The manuscript was improved according to reviewer's comments. 

Author Response

We thank the reviewer for giving detailed comments and suggestions that have been helpful to improve the manuscript.

Reviewer 2 Report

What is the exact study power based on your current sample size?

Author Response

1. We appreciate your comments and efforts in reviewing this manuscript.  We once again revised the manuscript for the language and style. 

2. Answer for the reviewer's comments: 

We used the R4.1.2, powerSurvEpi package to perform the power calculation based on the overall survival difference and sample size. The power in overall survival difference was 99.93%. The exact statistical power for the multivariate logistic regression models was 99.96%. We used Chi-Square Goodness of Fit Test in SPSS statistics and the subsequent data analysis for the statistical power was generated using the Real Statistic Resource Pack software (Release 7.6).

This manuscript is a resubmission of an earlier submission. The following is a list of the peer review reports and author responses from that submission.

Round 1

Reviewer 1 Report

Because there are contradictory results about the benefit of SCS, the authors evaluated the clinical characteristics of 262 patients with recurrent ovarian cancers to assess the impact of SCS on patient’s survival. They divided patients into two groups of SCS and chemotherapy alone. When they identified the most important factor to this different survival interval, they used multivariate analysis to efficiently control multiple confounding factors. Limited regional carcinomatosis was the only significant factor. They also analyzed their results based on the previous other clinical criteria. Their results were clear and the conclusions were reasonable. Some minor typos need to be modified.

Minor comments

1.     In Simple Summary (line 8), SCS was firstly used without full description. And the font of ‘Eligibility’ seemed to be different to others.

2.     In Results, line 123-124, the numbers were different to Table 1. Most patients (87.3%) vs. 87.4% in Table 1. 72% of the patients vs. 72.1% in Table 1.

3.     In Figure 1, the description of two groups and numbers are too small to read. The size need to be increased to make it readable.

4.     In Table 2, there is PFI instead of PFS. However, in full description for Table 2 abbreviation, there is explanation about PFS. They used both PFI and PFS without clear explanation. If these two words have the same meaning, it would be better to use only one word. But they were different; they should first explain them clearly, before using them.

Reviewer 2 Report

The study by Joo-Hyuk Son et al. developed simplified selection criteria for the secondary cytoreductive surgery in recurrent ovarian cancer. This is an interesting study and the context is important in the treatment of recurrent ovarian cancer. Grammatical revisions are needed to improve the quality of this manuscript.

My main concern is that the patient informed consent was waived. For studies involving human participants, informed consent is required. If patients biological samples would be used in future studies, a relevant statement should be added in the informed consent before collecting specimens for the use other than diagnosis and treatment. The following are my other comments;

Introduction:

1.     Line 48-49, explain more clearly what are the strictly defined criteria in DESKTOP and SOC-1 trial.

Materials and methods:

1.     A study flowchart with numbers of participants who were included and excluded at each phase is recommended to better describe the population selection process.

2.     Please clearly indicate the inclusion/ exclusion criteria for participants.

3.     Line 86-89 “In patients with …”. Please revise the English of this sentence. With all “previously know factors” or any one of these factors? Or a combination of factors?

4.     Line 89-90 “In general, patients with SCS have good performance status with no or minimal ascites.” does not fit in this context.

5.     Line 93-94, please explain clearly the definition of SCS. “At some time remote from…” is not precise. And a citation can be added here.

6.     Line 97 “following pretreatment …” please revise English of the sentence. It is very confusing.

7.     Line 107, substitute official with standard.

8.     A statistical power calculation is recommended based on the sample size.

Results:

1.     Clear subtitles of the results are recommended.

2.     Line 123. The percentage (87.3%) in the main text is not consistent with Table 1 (87.4%).

3.     P value is recommended to be added in Figure 1.

Discussion:

1.     More comparisons should be conducted with other studies, not just repeat what has already presented in the Introduction.

2.     Line 198-201 “In this study, about 25% of all patients in the surgery group had unresectable extra-abdominal disease and received SCS after second- to fourth-line chemotherapy. Although the patients were not suitable for the SCS at the time of first recurrence, they were included in eligible criteria (PFI > 12 months, limited carcinomatosis) after the chemotherapy.” More explainations are needed here. For example, how would this affect your results and its indications?

3.     Line 202-203 “The complete resection rate was 78.3% (18/23) and the second median PFS of this group after the SCS was 13 months (95% CI, 8.0–17.9 months).” Please clarify this sentence. Which group?

4.     Line 203-207 is not closely associated with the following conclusion “These findings suggest that if chemotherapies are adopted properly in the selection, the inclusion rate can be widened in patients not eligible at the time of first recurrence or patients accompanied with extra-abdominal diseases”.

5.     The selection bias should be discussed more fully, how would it affect your results? Towards or away from the null?